# Abundant Species Diversity and Essential Functions of Bacterial Communities Associated with Dinoflagellates as Revealed from Metabarcoding Sequencing for Laboratory-Raised Clonal Cultures

**DOI:** 10.3390/ijerph19084446

**Published:** 2022-04-07

**Authors:** Yunyan Deng, Kui Wang, Zhangxi Hu, Ying-Zhong Tang

**Affiliations:** 1CAS Key Laboratory of Marine Ecology and Environmental Sciences, Institute of Oceanology, Chinese Academy of Sciences, Qingdao 266071, China; yunyandeng@qdio.ac.cn (Y.D.); zhu@qdio.ac.cn (Z.H.); 2Laboratory of Marine Ecology and Environmental Science, Qingdao National Laboratory for Marine Science and Technology, Qingdao 266237, China; 3Center for Ocean Mega-Science, Chinese Academy of Sciences, Qingdao 266071, China; 4Institute for Advanced Study, Shenzhen University, Shenzhen 518060, China; kuiwang@szu.edu.cn

**Keywords:** Actinobacteria, algae-associated bacterial community, dinoflagellate, harmful algae, methylotrophs

## Abstract

Interactions between algae and bacteria represent an important inter-organism association in aquatic environments, which often have cascading bottom-up influences on ecosystem-scale processes. Despite the increasing recognition of linkages between bacterioplankton and dynamics of dinoflagellate blooms in the field, knowledge about the forms and functions of dinoflagellate-bacteria associations remains elusive, mainly due to the ephemeral and variable conditions in the field. In this study, we characterized the bacterial community associated with laboratory cultures of 144 harmful algal strains, including 130 dinoflagellates (covering all major taxonomic orders of dinoflagellates) and 14 non-dinoflagellates, via high-throughput sequencing for 16S rRNA gene amplicons. A total of 4577 features belonging to bacteria kingdom comprising of 24 phyla, 55 classes, 134 orders, 273 families, 716 genera, and 1104 species were recovered from the algal culture collection, and 3 phyla (Proteobacteria, Bacteroidetes, and Firmicutes) were universally present in all the culture samples. Bacterial communities in dinoflagellates cultures exhibited remarkable conservation across different algal strains, which were dominated by a relatively small number of taxa, most notably the *γ*-proteobacteria *Methylophaga*, *Marinobacter* and *Alteromonas*. Although the bacterial community composition between dinoflagellates and non-dinoflagellate groups did not show significant difference in general, dinoflagellates harbored a large number of unique features (up to 3811) with relatively low individual abundance and enriched in the potential methylotrophs *Methylophaga.* While the bacterial assemblages associated with thecate and athecate dinoflagellates displayed no general difference in species composition and functional groups, athecate dinoflagellates appeared to accommodate more aerobic cellulolytic members of Actinobacteria, implying a more possible reliance on cellulose utilization as energy source. The extensive co-occurrence discovered here implied that the relationships between these algal species and the bacterial consortia could be viewed as either bilaterally beneficial (i.e., mutualism) or unilaterally beneficial at least to one party but virtually harmless to the other party (i.e., commensalism), whereas both scenarios support a long-term and stable co-existence rather than an exclusion of one or the other. Our results demonstrated that dinoflagellates-associated bacterial communities were similar in composition, with enrichment of potential uncultured methylotrophs to one-carbon compounds. This work enriches the knowledge about the fundamental functions of bacteria consortia associated with the phycospheres of dinoflagellates and other HABs-forming microalgae.

## 1. Introduction

In natural aquatic ecosystems, algae grow in close association with the bacterial community, which forms an intrinsic component of algal physiology and ecology [1]. Amassed data show that the interactions between the bacterial associates and algal hosts are ubiquitous in both freshwater and marine systems [2,3]. Tight associations between algae and bacteria have been leading to the evolution of a complex network of cross-kingdom interactions and a fine specialization of different groups [3,4]. These interactions are mediated by diverse molecules and recognition mechanisms, which are generally categorized into three types: nutrient exchange, signal transduction, and horizontal gene transfer [5]. The exchange of metabolites and infochemicals results in distinct derived algae–bacteria relationships, varying from highly specific symbiont/host relationships to commensal/mutualist relationships or parasitic/algicidal behavior, to less-specific interactions such as nutrient competition/modification [2,3,6,7,8,9]. The interactions between these two groups either directly or indirectly affect the physiology of both partners, impact aquatic communities, alter ecosystem diversity, and influence global biogeochemical cycles [2,3,4,5,9]. 

The interactions between algae and bacteria primarily occur within the “phycosphere”, the aquatic analog of the rhizosphere in terrestrial plants, referring to the microenvironment surrounding algae in which gradients of released molecules support and enhance distinct bacterial communities [10]. Although taking place within an inherently microscale context, algae and bacteria interactions may often have cascading bottom-up influences on ecosystem-scale processes [2,3,5]. A better understanding of the mechanisms underlying their interactions and both partners that mediate their relationships are therefore essential and fundamental to predict changes in the aquatic ecosystems. A prerequisite for the association/exchange relationships is the spatial assembly of interacting species. Therefore, it is advisable to first categorize the taxonomic identities and abundances of the bacterial taxa that associate with different lineages of algal hosts [3,8]. Extensive studies have demonstrated specific combinations of algae/bacteria in phycospheres, suggesting the presence of specific interactions [4,5,11]. However, in marine habitats, these interactions are usually difficult to accurately explore, partly due to majority of photosynthetic algae being micro- or unicellular phytoplankton; thus, it is difficult to distinguish them from the associated free-living bacteria assemblages [2,3]. 

Dinoflagellates are the second largest phytoplankton linage and a major contributor to the primary productivity, food web, global carbon fixation, element cycle, and the balance of ecosystem energy flux [12]. Many species in this lineage are also notorious for being the most causative agents of harmful algal blooms (HABs), accounting for nearly 40% of the HABs-causing species worldwide [13]. HABs of dinoflagellates threaten coastal ecosystems, mariculture, tourism, as well as wildlife and human health through releasing toxins and triggering local oxygen depletion [14]. Though several studies have, via metagenomic sequencing of dinoflagellate blooms, documented that dinoflagellates–bacteria interactions have the potential to dramatically influence population dynamics [15,16,17], species-level information about the bacterial consortia characteristically associated with dinoflagellates still remains obscure. Members of the *γ*-proteobacteria *Marinobacter* and *Alteromonas* clades were documented as the dominant phylotypes associated with *Pfiesteria* sp. [18], *Alexandrium tamarense* [19], *Scrippsiella trochoidea* [19], *Noctiluca scintillans* [20], *A. fundyense* [21], *Alexandrium* sp. [11], *Gymnodinium catenatum* [22,23], and *Margalefidinium polykrikoides* [24] cultures. Bacteria belonging to *α*-proteobacteria *Roseobacter* were observed to co-exist with several dinoflagellates, including *Pfiesteria* sp. [18], *Scrippsiella trochoidea* [19], *Alexandrium tamarense* [19,24], *Alexandrium* sp. [11], *Gymnodinium catenatum* [22], and *Margalefidinium polykrikoides* (= *Cochlodinium polykrikoides*) [25] cultures. The majority of the abovementioned culture-based studies were anchored on two paralytic shellfish toxins (PSTs) producers, *A. tamarense* and *G. catenatum*, and centered on investigating the relationship between the associated bacterial flora and PST production. Moreover, these laboratory data mainly stemmed from cultivation of bacterial isolates from dinoflagellate cultures. Due to the striking number that potentially about 1000 bacterial genotypes have been reported as “attachments” to clonal cultures [18,19,23], and the bias in bacterial cultivation and inconsistency and persistence of bacterial assemblages across different dinoflagellate species [8], it is reasonable to state that our current knowledge about the species diversity and the variety of functions and associations of bacterial consortia around dinoflagellate cells is still fragmentary and, therefore, more comprehensive, intensive, and fundamental investigations are highly desirable. 

To gain more insights into the nature or form of the association and interaction between bacterial assemblages and dinoflagellates and other groups of microalgae, using the approach of high-throughput amplicon sequencing of 16S rRNA gene, we characterized the bacterial assemblages associated with 144 clonal cultures of harmful algae that have been established and cultured in our laboratory for from a few months to more than 7 years, including 130 strains of dinoflagellates (covering all major taxonomic orders of dinoflagellates) and 14 strains from other classes (categorized as “non-dinoflagellate” group). The objectives of this study were thus to investigate the diversity and composition of bacterial communities that may have been specifically associated with these microalgae and to characterize the possible functional profiles of these bacterial associations. The results obtained are believed to provide insightful understanding of the species composition and community functional profiles of dinoflagellate-associated bacterial assemblages. 

## 2. Materials and Methods

### 2.1. Strain Information, Grouping Details, and Sample Collection

A total of 144 strains of clonal cultures of microalgae were employed in this study, including 130 dinoflagellates (with 24 species having multiple strains) and 14 non-dinoflagellates (including 3 chlorophytes, 4 pelagophytes, 4 ochrophytes, 2 haptophytes, and 1 cryptophyte) (Appendix A). The identities of all cultures were confirmed by sequencing by high-throughput amplicon sequencing of 28S rRNA gene (~500 bp, covering the highly variable D2 domain and parts of the more conservative D1 and D3 domains). All the strains were divided into two groups, “DINO” (dinoflagellates) and “N-DINO” (non-dinoflagellates), and the DINO group was further categorized into Thecate (the 37 strains of armored species, i.e., with thecates) and Athecate (the 93 strains of naked species) groups. Moreover, 84 of the 144 strains that have been cultured with the addition of an antibiotics cocktail were labeled as “Anti” group. A final concentration of 2–3% (a penicillin-streptomycin mixture, 100×, Solarbio, Beijing, China) were added into the medium immediately before each time of the routine culture transfer for more than 12 months. The remaining 60 strains without antibiotics addition were labeled as “N-Anti” group (Appendix A).

All the algal strains were cultured in sterile filtered seawater (salinity of 32–33) enriched with f/2-Si culture medium ingredients under the same conditions as the routine laboratory maintenance of microalgal cultures: 20 ± 1 °C, 12:12 h light: dark cycle, and a photon flux density of ~100 μmol photons m^−2^·s^−1^ (cool white fluorescent lights). Cultures were kept in six-well culture plates (Corning, NY, USA) containing 10 mL of the seawater-based f/2-Si medium [26] in each well. The day of inoculation was recorded as Day 0. Vegetative cells were sampled on Day 15 when all cultures were at their stationary growth stage as pre-determined. All cells in each sample (approximately 10^4^~10^5^ cells) were pelleted in a 1.5 mL centrifuge tube and immediately used for total DNA extraction.

### 2.2. DNA Extraction and PCR Amplification

Genomic DNA of each sample was extracted with the Plant DNA Extraction Kit (Tiangen, Beijing, China) according to the manufacturer’s protocols. Total DNA was eluted with 50 μL TE buffer (Tris-hydrochloride buffer, pH 8.0, containing 1.0 mM EDTA). The DNA concentration and purity were determined spectroscopically using NanoDropTM 1000 spectrophotometer (Thermo Fisher Scientific, Waltham, MA, USA), then stored at −80 °C until PCR amplification.

The V3-V4 region of bacterial 16S ribosomal RNA gene was PCR-amplified (98 °C for 30 s; followed by 35 cycles at 98 °C for 10 s, 54 °C for 30 s, and 72 °C for 45 s; and a final extension at 72 °C for 10 min) using a primer set of 341F (5’-CCTACGGGNGGCWGCAG-3’) and 805R (5’-GACTACHVGGGTATCTAATCC-3’) [27]. The 5′ ends of the primers were tagged with specific barcodes per sample and sequenced using universal primers. All PCR reactions were conducted in a 25 μL mixture containing 12.5 μL of 2× Phusion^®^ Hot Start Flex Master Mix, 2.5 μL of each primer (1 μM), and 50 ng of template DNA. Nuclease-free water served as blank.

### 2.3. Illumina MiSeq Sequencing

The PCR products were confirmed with 2% agarose gel electrophoresis. Then the amplicons were extracted from agarose gels and purified with the AxyPrep DNA Gel Extraction Kit (Axygen Biosciences, Union City, CA, USA) according to the manufacturer’s instructions. The size and quantity of the purified amplicon library were assessed on Agilent 2100 Bioanalyzer (Agilent, Santa Clara, CA, USA). The library was sequenced on the NovaSeq PE250 platform at LC-Bio Technology Company (Hangzhou, China).

### 2.4. Sequencing Data Processing and Bioinformatic Analyses

Paired-end reads was assigned to samples based on their unique barcode, respectively, then truncated by cutting off the barcode and primer sequence, and merged using FLASH [28]. Quality trimming and length filtering were performed on the raw reads in Fqtrim software, and the chimeric reads were further filtered using Vsearch software [29]. The amplicon sequence variants (ASVs) were generated with DADA2 package [30]. All the features were annotated by conducting BLAST search against the SILVA database [31]. The relative abundance of each taxon was computed by normalizing the number of assigned reads to the number of total reads sequenced, and the average abundance of each group was calculated. Rarefaction curves were generated for each sample using custom Perl scripts. Venn diagrams showing the shared and unique features were plotted with BioVenn (http://www.biovenn.nl/index.php accessed on 2 January 2022). The alpha diversity indices (Chao1 richness, Observed species, Goods coverage, Shannon diversity, and Simpson evenness) were calculated via QIIME 2 (quantitative insights into microbial ecology 2) to analyze the complexity of species diversity [32]. The significance of variance between or among samples was tested with one-way ANOVA or *t*-test (for comparison between two groups) using the software SPSS 22.0 (SAS Institute Inc., Cary, NC, USA). Beta diversity analyses were performed to display and compare bacterial community composition among different groups. PCA (principal component analysis) was conducted with QIIME 2 plugin [32]. The PCoA (principal coordinated analysis) and ANOSIM (analysis of similarities) were conducted based on the weighted−uniFrac distance. An ordination plot was produced by NMDS (non-metric multidimensional scaling) using the ranked similarity matrix, generated with PRIMER program [33]. Hierarchical agglomerative clustering (to group objects in clusters based on their similarity) using the group average method was carried out on the most abundant features according to groups selected from NMDS analysis. The LDA (Linear discriminant analysis) effect size analysis (LEfSe) was performed using the OmicStudio tools to determine the observed features that most likely to explain differences between samples by coupling standard tests for statistical significance with additional tests encoding biological consistency and effect relevance [34]. A heatmap of bacterial communities was generated using the PHYLOTEMP tool, with relative abundance data clustered based on the Bray-Curtis similarity algorithm [35]. The significance level in all statistical analyses was set at 0.05 unless otherwise stated.

### 2.5. Functional Annotation of the Presented Common Bacterial Communities

To explore potential functional differences among the bacterial communities between different groups, the metagenome for each group was predicted by using PICRUSt2 (Phylogenetic Investigation of Communities by Reconstruction of Unobserved States) algorithm [36]. Bacterial gene functions were predicted from 16S rRNA gene-based microbial species compositions using the PICRUSt algorithm to make inferences from KEGG database [37]. The differentially abundant gene families and pathways were assessed using the software STAMP [38] subjected to *t*-test and Tukey–Kramer post hoc analysis with Benjamini-Hochberg FDR multiple comparison correction. A significant difference was inferred when *p* < 0.05.

## 3. Results

### 3.1. Global Overview of Bacterial Diversity and Community Composition Associated with Algal Cells

A total of 12,022,991 raw rDNA sequence reads, corresponding to 5.87 Gb of raw data, were obtained from the 144 samples, with an average of 83,493 sequences per sample. The raw sequencing data were deposited in the NCBI Short Read Archive (SRA) database with the accession number PRJNA771505. After removing short sequences, poor quality sequences, and chimeric sequences, 10,635,295 effective sequences (approximately 4.37 Gb) remained, with a quality control efficiency of 88.46% (Appendix A). Upon dereplication using DADA2 within the QIIME2 tool, the data set finally yielded 4738 prokaryotic features for subsequent analyses. Goods coverage of all samples were 1.00 (Appendix A) and all rarefaction curves tended to reach saturation with increased sequencing amounts (Appendix A), indicating sufficient sequencing depth for revealing the species diversity of prokaryotic taxa associated with the algal culture samples.

All the 4738 effective prokaryotic features were further assigned to different taxonomical levels from kingdom to species via blasting against the database Silva (https://www.arb-silva.de/ accessed on 23 April 2021). Features annotated as “unclassified” kingdom were excluded from the following analyses. The remaining 4577 features belonging to the bacteria kingdom were assigned to 24 phyla, 55 classes, 134 orders, 273 families, 716 genera, and 1104 species (Appendix A). The number of features per culture varied from 44 to 592 (mean = 101) among the 144 samples (Appendix A). The most abundant microbial phylum was Proteobacteria (91.49%), and other major groups were Bacteroidetes (3.34%), Cyanobacteria (3.17%), and Planctomycetes (1.16%) (Figure 1a and Appendix A). At the genus level, the top 30 in abundance are shown in Figure 1b, with Methylophaga (20.33%), Marinobacter (17.77%), Alteromonas (12.42%), and Alcanivorax (8.15%) being the 4 most abundant genera, which all together account for 58.7% of all features (Appendix A). 

### 3.2. Species Composition of Bacterial Communities Associated with DINOs and N-DINOs and their Predicted Functions

Comparing the DINO (130 dinoflagellate strains) and N-DINO (14 non-dinoflagellate strains) groups, no significant difference was detected in the alpha diversity indices (Shannon diversity, Simpson evenness, and Observed species) (ANOVA, *p* > 0.05; Figure 2a). Further beta diversity analyses were performed to visualize similarity and dissimilarity between groups in species (features; 100% identity) complexity. Both PCA and PCoA plots based on the weighted−uniFrac distance revealed that the group DINO was not distinct from N-DINO (Figure 3a,b). NMDS is another important index of beta diversity, in which result is evaluated by stress coefficient. Our NMDS plot using the Bray-Curtis similarity method showed the stress coefficient was 0.16, indicating there was no significant difference (*p* > 0.05) in bacterial species composition between DINO and N-DINO (Figure 3c).

For the differences in bacterial features between the two groups, as shown in a Venn diagram (Figure 2b), while two groups shared 458 bacterial features in common, the DINO and N-DINO group contained 3811 and 469 unique features, respectively. At the phylum level, among the 24 bacterial phyla identified from the two groups, Proteobacteria, Bacteroidetes, and Planctomycetes were found to be the “big three” predominant phyla (relative abundance > 1%) (Figure 4a). The Proteobacteria phylum richness in the DINO group was significantly higher than that in the N-DINO group, but the relative abundance of Cyanobacteria and Chlamydiae was dramatically higher in N-DINO group (Figure 4a and Appendix A). At genus level, *Methylophaga*, *Marinobacter*, *Alteromonas*, *Alcanivorax*, *Ponticoccus*, *Thalassospira*, and *Roseovarius* were the most dominant genera for all samples as a whole (relative abundance of each >1%) (Figure 4b). Compared with the N-DINO group, the DINO group significantly higher relative abundances of *Methylophaga*, *Erythrobacter*, *Phaeomarinobacter*, *Bifidobacterium*, *Salinimonas*, *Burkholderia*, and *Faecalibacterium* detected, but significantly lower relative abundances of *Oxyphotobacteria*, *Stenotrophomonas*, *Roseibium*, *Neisseria*, *Rhodopirellula*, *Corynebacterium 1*, *Rothia*, *Labrenzia*, *Haemophilus*, *Hyphomonas*, *Ponticoccus*, *Alistipes*, unclassified genera of Erythrobacteraceae, and Phascolarctobacterium (Figure 4b and Appendix A). LEfSe analysis indicated that *Methylophaga*, *Roseovarius*, Cryomorphaceae unclassified, Croceitalea, Simkaniaceae unclassified, Oxyphotobacteria unclassified, Lachnospiraceae XPB1014 group, *Ruminococcus*, *Mitsuokella*, OM190 group, Planctomycetales unclassified, *Labrenzia*, *Ponticoccus*, Erythrobacteraceae unclassified were discriminating taxonomic units between dinoflagellates and non-dinoflagellates (Figure 5).

To explore the possible ecological roles played by the phycosphere bacteria, functional predictions were performed using the PICRUSt algorithm. Three categories, genetic information processing, unclassified, and cellular processes, showed significantly higher activity in DINO than in the N-DINO group, whereas organismal systems and metabolism were dramatically over-represented in the N-DINO group (KEGG level 1; Figure 6a). For the secondary functional modules (KEGG level 2), 10 categories of functions (replication and repair, transcription, genetic information processing, poorly characterized, glycan biosynthesis and metabolism, biosynthesis of other secondary metabolites, cellular processes and signaling, transport and catabolism, signal transduction and cell motility) were markedly enriched in the DINO group. N-DINO had significantly more relative abundances of metabolism of membrane transport, metabolism of cofactors and vitamins, translation, carbohydrate metabolism, metabolism of other amino acids, enzyme families, energy metabolism, signaling molecules and interaction, amino acid metabolism, folding, sorting and degradation, and lipid metabolism (Figure 6b). Differential function prediction at KO (KEGG Orthology) assignments also revealed members being potentially involved in methanogenesis, including methenyltetrahydrometh-anopterin cyclohydrolase, methylene-tetrahydromethanopterin dehydrogenase, formylmethanofuran-tetrahydromethanopterin formyltransferase, tetrahydrometh-anopterin hydrolyase, and formylmethanofuran dehydrogenase subunit A, B, C, were predominantly higher in the DINO group than in the N-DINO group (Appendix A).

### 3.3. Comparative Analysis of the Species Composition and Predicted Function of Bacterial Communities between the Thecate and Athecate Groups

The 130 dinoflagellates were grouped into Thecate (37 strains of the armored) and Athecate (93 strains of the naked) groups. Overall, the two groups shared 716 common bacterial features (17.2 %, as total of 4269 OTUs), while Thecate and Athecate groups had 615 and 2938 unique features, respectively (Figure 2b). At the phylum level, significantly higher relative abundances of Cyanobacteria, Patescibacteria and Actinobacteria were found in the Athecate group than in the Thecate group; whereas more Verrucomicrobia and Deferribacteres members were present in the Thecate group (Figure 7a). Subsequently, bacterial classes of Oxyphotobacteria, Planctomycetacia, Negativicutes, Saccharimonadia, and Actinobacteria exhibited higher abundance in the Athecate group, whereas Verrucomicrobiae, Deltaproteobacteria, and Deferribacteres were significantly higher in the Thecate group (Figure 7b). At genus level, the Athecate bacterial communities were notably enriched with *Oxyphotobacteria*, *Lautropia*, *Nonlabens*, *Mesorhizobium*, *Klebsiella*, *Methylotenera*, *Reichenbachiella*, *Porphyromonas*, *Roseovarius*, *Bacteroides*, *Blastopirellula*, and unclassified Actinobacteria and Gammaproteobacteria members (Appendix A). In contrast, the Athecate group had higher relative abundances of *Coprococcus*, Lachnospiraceae UCG-006, *Tyzzerella*, *Labrenzia*, *Roseburia*, *Akkermansia*, *Thalassobius*, *Duncaniella,* Lachnospiraceae NK4A136 group, as well as those unclassified Phycisphaeraceae, Nannocystaceae, Muribaculaceae, and Rhodobacterales genera (Appendix A). Nonetheless, no significant difference in bacterial communities between Thecate and Athecate groups could be solved by alpha or beta diversity analyses (Appendix A).

Regarding the predicted putative functions of bacterial communities, no significant difference was detected between the Thecate and Athecate groups at KEGG level 1. At KEGG level 2, the category, signaling molecules, and interaction, showed a markedly higher relative abundance in the Thecate group (Figure 8a). At KEGG level 3 (functional modules), 9 categories (signal transduction mechanisms, amino acid metabolism, mismatch repair, others, pentose and glucuronate interconversions, protein export, polycyclic aromatic hydrocarbon degradation, cellular antigens, and histidine metabolism) were significantly enriched in the Thecate group (Figure 8b). The Athecate group was more enriched with other 8 categories of metabolism: vitamin B6 metabolism, general function prediction only, basal transcription factors, bacterial secretion system, calcium signaling pathway, protein folding and associated processing, RNA transport, and transcription related proteins (Figure 8b). Furthermore, we found predominant enrichment of KOs pertaining to the transport system of cellobiose in the Athecate group than those in the Thecate group, which included cellobiose transport system substrate-binding protein, cellobiose transport system permease protein cebF and cellobiose transport system permease protein cebG (Appendix A).

### 3.4. Comparative analysis of the species composition and predicted function of bacterial communities between the Anti and N-Anti groups

Comparing the two groups with (Anti group, 84 strains) and without (N-Anti, 60 strains) continual additions of the antibiotics mixture (penicillin and streptomycin), no significant difference in alpha and beta diversity of bacterial communities was found (Appendix A). The Venn diagram showed that 776 bacterial features were shared by both groups, whereas 1824 and 2117 unique features were present in the Anti and N-Anti groups, respectively (Figure 2b). At phylum level, the richness of Synergistetes, Acidobacteria, Firmicutes, and Thermotogae in Anti group were significantly lower than those in the N-Anti group (Figure 9a). The abundances of two phyla, Planctomycetes and Cyanobacteria, were found to be significantly elevated in the Anti group (Figure 9a). Among 97 bacterial genera detected with significant differences between the two groups, 63 genera had higher abundance in the Anti group, whereas 34 genera were more abundant in the N-Anti group (Figure 9b and Appendix A).

Differences in the functional potential of bacterial communities were found between the Anti and N-Anti groups. A significantly higher activity of genetic information processing and cellular processes, as well as a greatly lower activity of organismal systems, metabolism, and unclassified were detected in the N-Anti group at KEGG level 1 (Figure 10a). The categories of enzyme families, transcription, metabolism, metabolism of other amino acids, membrane transport, xenobiotics biodegradation and metabolism, amino acid metabolism, metabolism of cofactors and vitamins, glycan biosynthesis and metabolism, carbohydrate metabolism, lipid metabolism, cell growth and death, membrane transport, biosynthesis of other secondary metabolites, transport and catabolism, signaling molecules and interaction, metabolism of terpenoids and polyketides showed much higher relative abundance in the Anti group at KEGG level 2 (Figure 10b). Compared with the Anti group, the N-Anti group had dramatically elevated metabolism with respect to environmental adaptation, energy metabolism, cellular processes and signaling, genetic information processing, replication and repair, nucleotide metabolism, translation, signal transduction, cell motility, folding, sorting and degradation functions (Figure 10b).

## 4. Discussion

### 4.1. The Long-Lasting Bacterial Associations to Laboratory-Raised Algal Cultures Hints Bilaterally or/and Unilaterally Beneficial Relationships between Algae and Bacteria

Within the context of ecosystem function, the ecological relationships between phytoplankton and bacteria may represent the most important inter-organism association in aquatic environments [2,3]. Despite increasingly documented evidence of linkages between bacterioplankton and dynamics of dinoflagellate blooms in nature [15,16,17,39], only limited knowledge of dinoflagellates-characteristic/unique bacterial associations and their functional implications are available. In this study, based on the culture (cultivation of bacteria from algal cultures)-independent high-throughput 16S rRNA gene amplicon sequencing, we delineated the diversity and composition of bacterial associations with 144 strains of laboratory algal cultures. Generally, Proteobacteria categorically predominate (91.49%) the associate community, with Bacteroidetes as sub-dominant (3.34%), followed by Cyanobacteria (3.17%), and then Planctomycetes (1.16%) being relatively common (relative abundance > 1%). Much research has shown that complex and dynamic interactions between the associated bacterial community and phytoplankton are ubiquitous in aquatic systems [2,3]. The bacterial contribution and function in phytoplankton growth were demonstrated by culture-based studies and have been increasingly recognized in recent years [8,25]. Meanwhile, previous works on coastal plankton communities found that the phytoplankton production relied on metabolically active heterotrophic bacterial community even when sufficient inorganic nutrients were available, implying that the bacterioplankton is an essential factor for phytoplankton production in nature [40]. Furthermore, the dependence of phytoplankton on their associated bacterial community has been proposed as a unique mechanism of nutrient recycling and growth regulation [3,5,6,7]. Microalgae growth depends on bacteria providing source for assimilation of certain nutrients, such as nitrogen, iron, sulfur, and vitamin B_12_, indicating that their interaction is more complex and significant than expected [4,6,7]. In this study, all the 144 strains were maintained under routine laboratory conditions for more than 12 months, amongst one fifth of them had been isolated and cultured in the laboratory for more than 7 years. A total of 4577 features belonging to bacteria kingdom comprising of 24 phyla, 55 classes, 134 orders, 273 families, 716 genera, and 1104 species were recovered from the algal culture collection, while 3 phyla (Proteobacteria, Bacteroidetes, and Firmicutes) were universally present in all the culture samples. Therefore, logically, the extensive co-occurrence revealed here implied that the relationships between these algal species and the bacterial consortia should be viewed as either bilaterally beneficial (i.e., mutualism) or unilaterally beneficial at least to one party but virtually harmless to the other party (i.e., commensalism), but both scenarios support a long-term and stable co-existence rather than an exclusion of one or the other. 

### 4.2. Bacterial Communities of Dinoflagellates Display Strong Conservation across Strains with an Enrichment of Methylophaga from the Class γ-Proteobacteria and Implies a Potentially Functional Group of Methylotrophs in Methane Consumption

Neither alpha nor beta diversity analyses could clearly differentiate DINO and N-DINO groups in species diversity and community composition of bacterial communities at the feature level (100% identity). This result was possibly owing to high nutrient and algal biomass, which may lead to high substrate availability for bacterial survival in the cultures [41], allowing for the same and/or affinitive bacteria to associate with different phytoplankton at similar physiological status. Although DINO and N-DINO groups did not show distinct community compositions, DINO group indeed harbored a large number of unique bacterial features (up to 3811) with a relatively low abundance. Such distinct associations may imply different metabolic potential for these bacterial taxa, but we are uncertain how these variations affect the bilateral interactions in these dinoflagellate strains.

Bacterial communities of dinoflagellates display strong conservation across strains, which were dominated by a relatively small number of taxa. Among them, the most notable (relative abundance > 10%) were members belonging to *γ*-proteobacterial clades of *Methylophaga* (24.25%), *Marinobacter* (21.18%), and *Alteromonas* (18.26). These results were in congruent with previous works, showing that members affiliated with *Marinobacter* and *Alteromonas* clades within the class *γ*-proteobacteria were the dominant phylotype in laboratory dinoflagellate cultures [11,18,19,20,21,22,23,24]. Intriguingly, we found that *Methylophaga* association with dinoflagellates has been rarely documented. Bacteria belonging to the genus *Methylophaga* are a unique group of aerobic, halophilic, non-methane-utilizing methylotrophs [42]. Shin et al. (2018) documented that the most abundant four genera associated with a laboratory-cultured *Margalefidinium polykrikoides* included *Methylophaga* [24]. Hattenrath-Lehmann et al. (2019) reported that *Methylophaga* (<0.5%) were present among field samples of *Margalefidinium polykrikoides* bloom in New York, USA. The *Methylophaga* members also have no cultured representative from dinoflagellate cultures [39]. However, in the present study, *Methylophaga* represented the most significantly enriched genus associated with dinoflagellates, relative to that associated with non-dinoflagellates. LEfSe analysis also supported *Methylophaga* being one of the discriminating genera between DINO and N-DINO groups. Dinoflagellates are recognized as the most prolific dimethylsulfoniopropionate (DMSP) producers [43], with intracellular concentrations sometimes exceeding 1–2 M [44]. As one of the earth’s most abundant organosulfur molecules, DMSP can degrade via DMSP cleavage to liberate the volatile sulfur compound dimethylsulfide (DMS) [44,45]. Different bacterial lineages have distinct capabilities and/or preference in terms of utilizing specific substances released from phytoplankton [46]. It is reported that the majority of marine bacteria capable of growth on DMS as a sole carbon source are members of the genus *Methylophaga* [47]. Therefore, the most likely explanation for the abundant and extensive presence of *Methylophaga* in phycospheres of DINO group is that the large amount of dinoflagellate-derived DMSP serves as the biogenic precursor for DMS production, which is utilized by *Methylophaga* as a source of carbon, sulfur, and/or energy.

In addition, we found predominant enrichment of KOs (KEGG Orthology) pertaining to methanogenesis in DINO group. All these predicted KOs, including methenyltetrahydromethanopterin cyclohydrolase, methylene-tetrahydromethanopterin dehydrogenase, formylmethanofuran-tetrahydromethanopterin formyltransferase, tetrahydromethanopterin hydrolyase, formylmethanofuran dehydrogenase subunit A, B, C, are involved in CO_2_ activation and reduction to methane [48,49]. Methane is a typical one-carbon (C1) source in marine environment. A number of marine methylotrophs have been enriched and isolated via using methane as sole carbon source in [42] and the references therein. Here, our results seemed to also imply that methylotrophic bacteria played a role in their consumption of methane in the DINO group. Knowledge of marine methylotrophs is largely based on enrichment and cultivation studies using defined media and high concentrations of substrate [42]. However, this kind of research has not yet been conducted for dinoflagellates. Therefore, this finding provides clues to further uncover an ecological adaptation of particular uncultured methylotrophs to methane, with a close relationship with dinoflagellates.

### 4.3. Athecate Dinoflagellates Provided a Better Niche for Aerobic Cellulolytic Bacteria from the Phylum Actinobacteria and a Possible Reliance on Cellulose Utilization as Energy Source

As described above, the 130 strains of dinoflagellate employed in this study were categorized into Athecate (93 strains) and Thecate groups. In general, bacterial communities associated with the two groups displayed a high similarity in species composition, function prediction, and their level 1 KEGG pathway functions predicted using PICRUSt2. Even at KEGG level 2, only one category of signaling molecules and interaction showed a significantly higher relative abundance in the Thecate group. These results indicate a high similarity in the chemical composition of the phycospheres of all dinoflagellates, which supports generally similar bacterial consortia. 

The cell surface and cell walls of dinoflagellates consist of a robust and intricate multilayered mesh of polysaccharides, proteins and other macromolecules, with celluloses being the major component (see [50] and the references therein). Some dinoflagellates have prominent membrane-bound thecal plates, which have high cellulose contents and possess mechanical properties similar to the cell wall of softwood [51]. Until now, cellulose hydrolysis and utilization were thought to be carried out exclusively by microorganisms [52,53]. In bacteria, cellulose utilization is predominantly found in the aerobic Actinomycetales of the phylum Actinobacteria and the anaerobic order Clostridiales belonging to the phylum Firmicutes [52,53,54]. In our study, it was notable that the Athecate group showed significantly stronger enrichment in Actinobacteria phyla and Actinobacteria class, implying more preference for cellulose utilization in this group. This speculation was also supported by differential function prediction at KO assignments, which showed the KOs (cellobiose transport system substrate-binding protein, cellobiose transport system permease protein cebF and cellobiose transport system permease protein cebG) potentially functioning in the transport system of cellobiose, the major hydrolytic product of cellulose, were predominantly higher in the Athecate group than those in the Thecate group. Celluloses in terrestrial plants share many characteristics across different lineages of taxa, including their potential for complete hydrolysis and utilization in sufficient amounts to provide energy to an organism under proper environmental conditions [52,54,55]. Two fundamentally different strategies for cellulose utilization are employed by the aerobic and anaerobic groups in cellulolytic bacteria [54,55]. Aerobic cellulolytic bacteria utilize cellulose through the production of substantial amounts of extracellular cellulase enzymes, which in turn promote to the high-cell-yield characteristic of aerobic respiratory growth [55]. Therefore, our results together seemed to hint that the bacterial community associated with the Athecate group may rely on cellulose utilization to provide energy. Any further, deeper understanding of the physiological properties of cellulolytic microorganisms is expected to shed light on several essential issues concerning cellulose-degrading communities, especially for their ecological implication.

### 4.4. Antibiotics Addition Does Not Significantly Affect Bacterial Diversity and Community Composition Associated with Our Microalgal Cultures

The 84 cultures of the Anti group have been treated with addition of a penicillin–streptomycin mixture for anywhere from 12 months to more than 7 years, while the remaining 60 strains of the N-Anti group have been cultured without antibiotics addition for anywhere from 12 months to more than 7 years (temporally equivalent to Anti group). Prokaryotes, however, were still ubiquitously present in all 144 cultures. Apparently, the addition of the antibiotics mixture was not successful in eliminating bacteria in our algal cultures, although it must have inhibited the overgrowth of bacteria, as evidenced by the healthy maintenance of our algal cultures. While a number of mechanisms may explain the bacteria assemblage surviving the continuous presence of antibiotics, such as the well-known adaptive persistence developed during low or sublethal dosages of antibiotics application [56,57,58], inherited resistance [57,59] to the types of antibiotics (penicillin and streptomycin) as used in this study, and even the protective “shields” provided via a sticky association between bacteria and algal cell surfaces [3,60]. Moreover, some bacteria are endosymbionts of dinoflagellates [61,62]; thus, they are not exposed to antibiotics. On the other hand, viewing this antibiotics-resistant association of bacteria to our algal cultures from the point of ecology of HABs in particular and of phytoplankton in general, algal cell-hosted bacteria must play more or less essential roles in the dynamics of microalgae.

## 5. Conclusions

Based on high-throughput amplicon sequencing of the 16S rRNA gene, we characterized the bacterial assemblages associated with 144 algal laboratory culture strains (most are HAB species). The co-occurrence of highly diversified bacteria implies that the association between these algal species and bacterial consortia is either bilaterally (i.e., mutualism) or at least unilaterally (i.e., commensalism) beneficial to the two partners. Neither alpha nor beta diversity analyses could disclose significant difference between DINO and N-DINO groups in species diversity and community composition of bacterial communities, which was possibly owing to the high nutrient levels and algal biomass leading to high bacterial substrate availability in the laboratory cultures, allowing the same and/or affinitive bacteria to associate with different phytoplankton at similar physiological statuses. Dinoflagellates generally harbored bacterial communities of strong conservation across different strains with exclusive enrichment of *Methylophaga* belonging to the class *γ*-proteobacteria and potential methylotrophs in their consumption of methane. While bacterial associations with Thecate and Athecate groups displayed compositional and functional similarities, the Athecate group showed a more preferred niche for aerobic cellulolytic members in Actinobacteria phyla, implying a plausible proneness to utilized cellulose as an energy source. Taken together, our results provided some important information and evidence towards a better understanding on the fundamental functions of bacteria consortia associated with the phycospheres of dinoflagellate and other HABs-forming microalgae.

## Figures and Tables

**Figure 1 ijerph-19-04446-f001:**
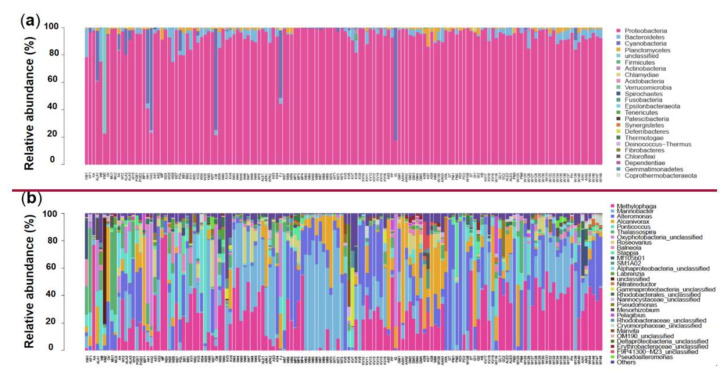
Relative abundance of different phyla (**a**) and genera (top 30; (**b**)) in the 144 samples. The abundance is presented in terms of percentage in total effective features in a sample. The order of sample ID on the X-axis is in accordance with that in the “Sample ID” column in Appendix A.

**Figure 2 ijerph-19-04446-f002:**
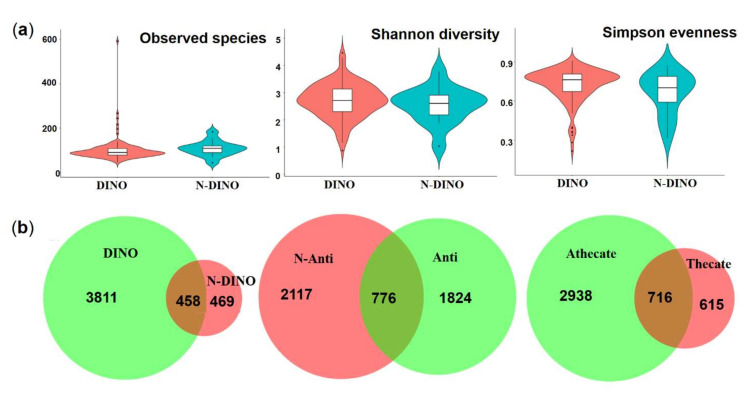
(**a**) Violin plots (median, min and max) showing alpha diversity (Shannon diversity, Simpson evenness and the number of observed species) of DINO (dinoflagellates) and N-DINO (non-dinoflagellates) groups. (**b**) Venn diagram showing the numbers of shared and unique bacterial features.

**Figure 3 ijerph-19-04446-f003:**
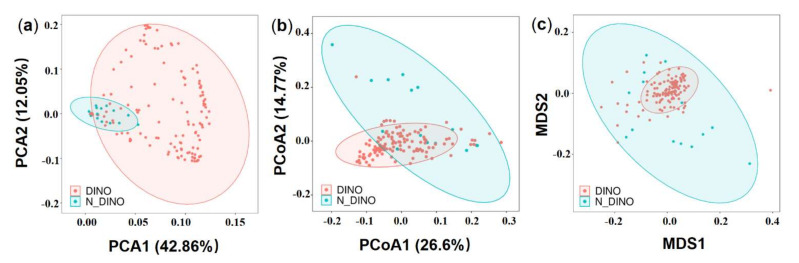
The beta diversity analysis of DINO (light red) and N-DINO (light blue) groups. (**a**) Principal component analysis (PCA) using genus/species-level Hellinger transformed relative abundances of bacterial sequences. (**b**) Principal coordinate analysis (PCoA) using weighted−unifrac distances. (**c**) Non-metric multidimensional scaling (NMDS) plot based on the weighted−unifrac distance using the Bray–Curtis similarity method.

**Figure 4 ijerph-19-04446-f004:**
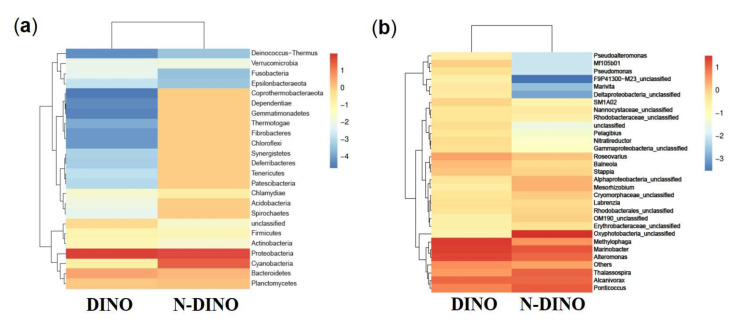
Heatmap of hierarchy cluster results representing the most abundant bacterial features at the phylum (**a**) and genus (**b**) levels in DINO and N-DINO groups.

**Figure 5 ijerph-19-04446-f005:**
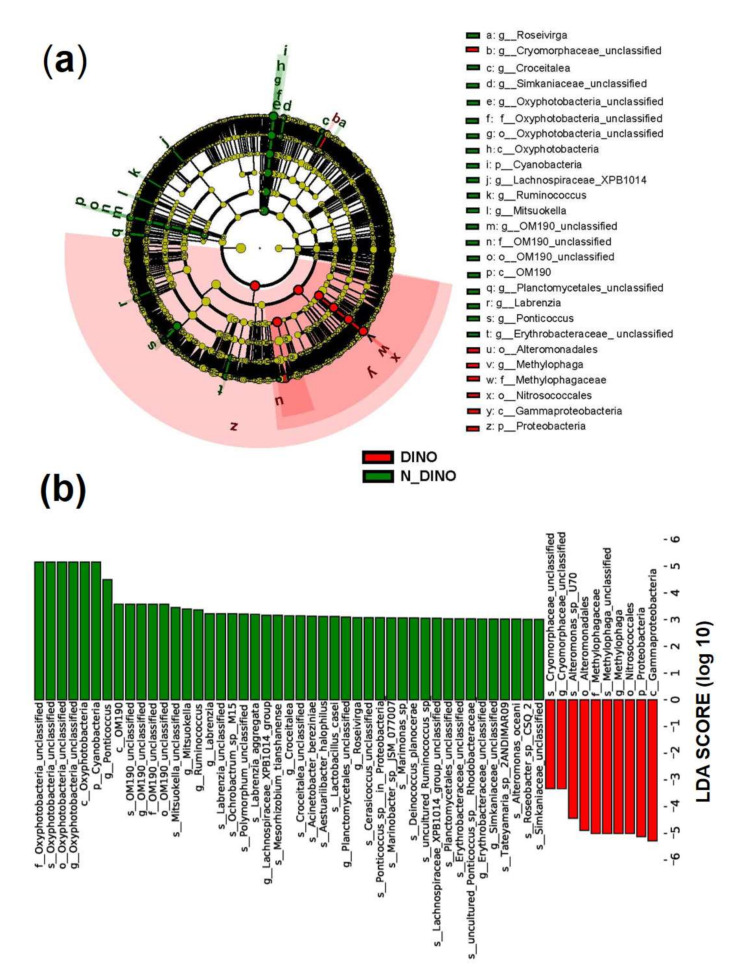
Linear discriminant analysis (LDA) integrated with effect size (LEfSe). (**a**) Cladogram illustrating the phylogenetic distribution of bacterial taxa differentially represented between DINO (red) and N-DINO (green) groups. (**b**) The differences in abundance of represented taxa between DINO (red) and N-DINO (green) groups.

**Figure 6 ijerph-19-04446-f006:**
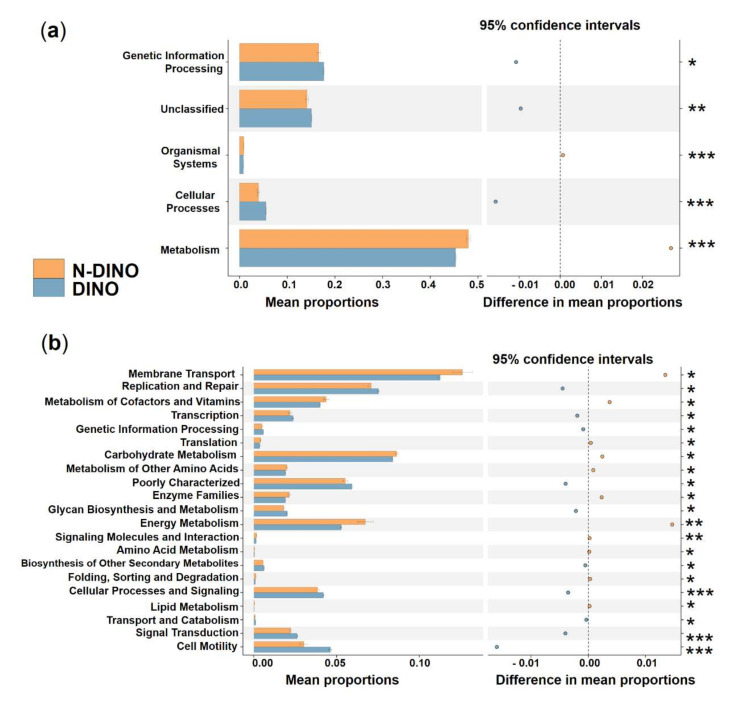
Prediction of the differential function of bacterial communities between DINO (blue haze) and N−DINO (orange) groups in KEGG categories at level 1 (**a**) and level 2 (**b**). Gene functions were predicted from 16S rRNA gene−based microbial compositions using the PICRUSt algorithm to make inferences from KEGG annotated databases. Relative signal intensity was normalized by the number of the genes for each indicated metabolic pathway. *, **, *** indicate the difference is at a significant level with *p* < 0.05, *p* < 0.01, and *p* < 0.001, respectively.

**Figure 7 ijerph-19-04446-f007:**
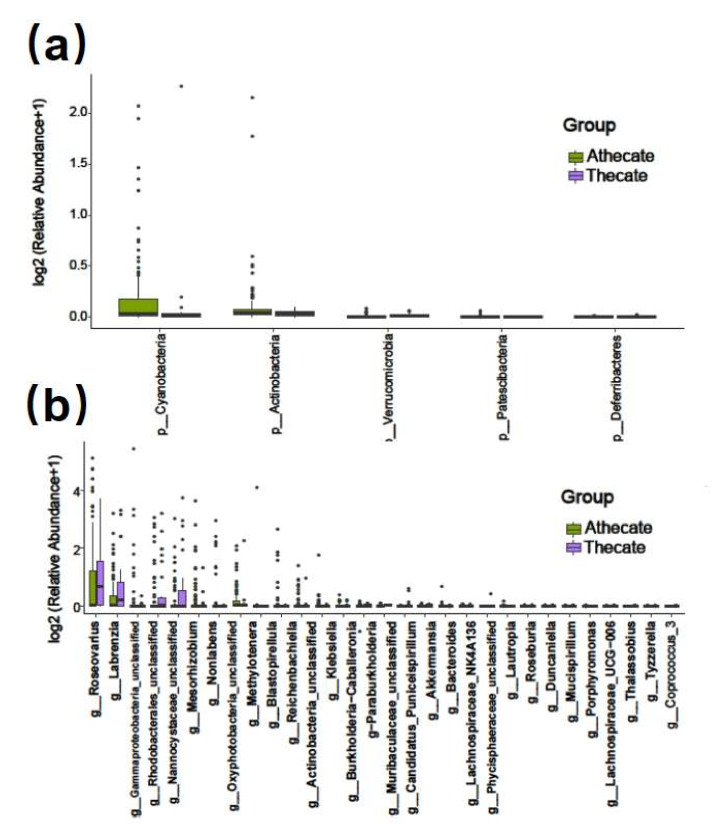
Bar plot of significantly different bacterial phyla (**a**) and classes (**b**) between Thecate and Athecate groups.

**Figure 8 ijerph-19-04446-f008:**
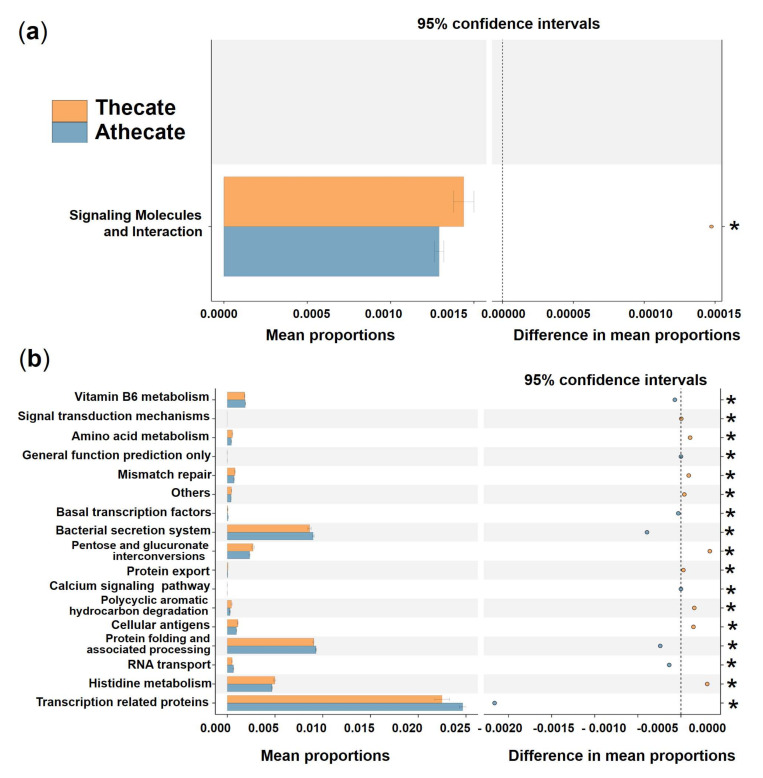
Prediction of the differential function of bacterial associations between the Thecate (orange) and Athecate (blue haze) groups in KEGG categories at level 2 (**a**) and level 3 (**b**). Gene functions were predicted from 16S rRNA gene-based microbial compositions using the PICRUSt algorithm to make inferences from KEGG annotated databases. Relative signal intensity was normalized by the number of the genes for each indicated metabolic pathway. * notes the difference is at a significant level with *p* < 0.05.

**Figure 9 ijerph-19-04446-f009:**
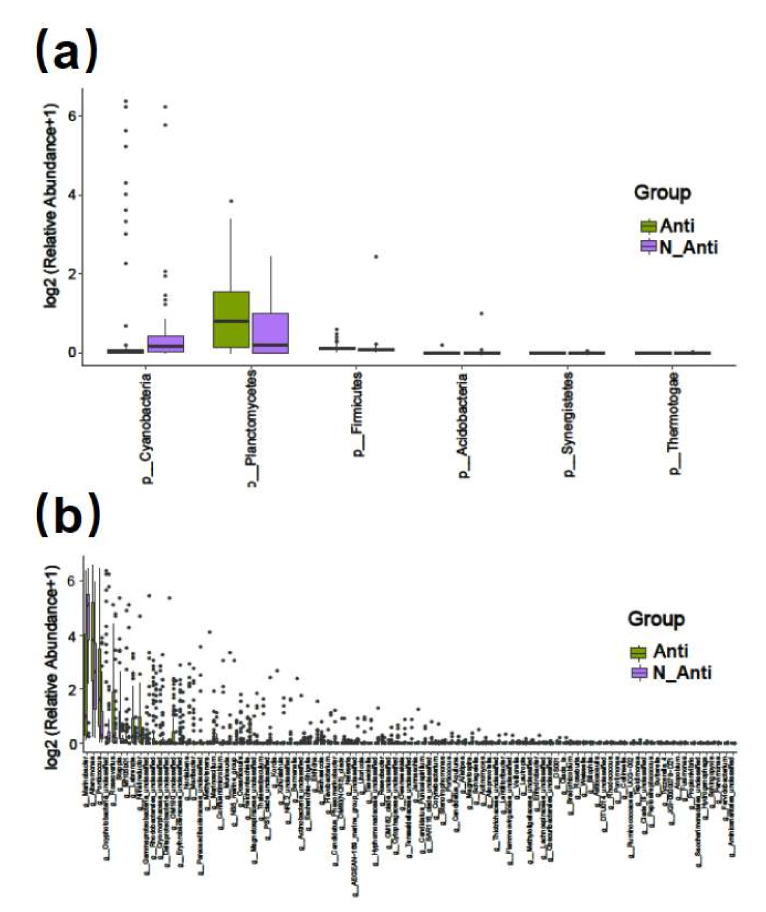
Bar plot of significantly different bacterial phyla (**a**) and genera (**b**) between the Anti and N-Anti groups.

**Figure 10 ijerph-19-04446-f010:**
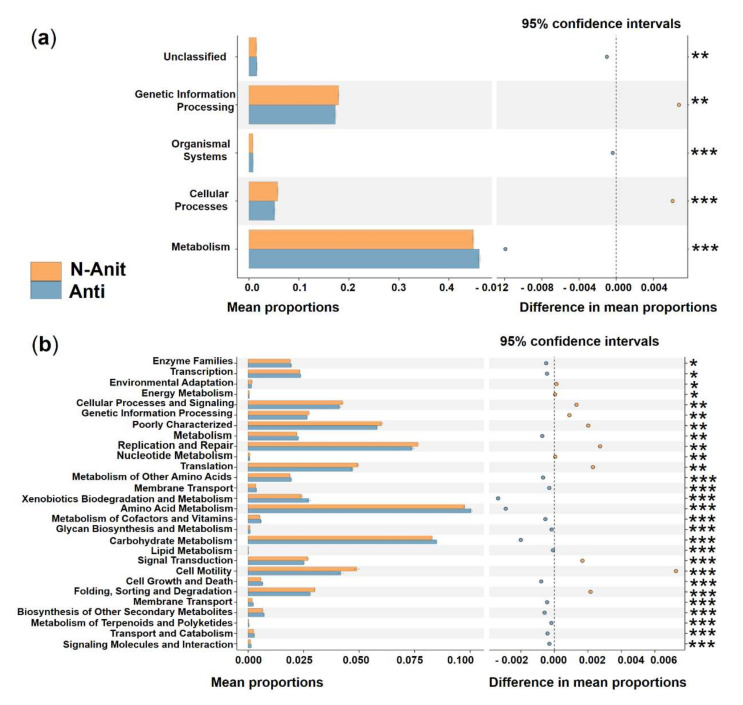
Prediction of the differential function of bacterial associations between the N−Anti (orange) and Anti (blue haze) groups in KEGG categories at level 1 (**a**) and level 2 (**b**). Gene functions were predicted from 16S rRNA gene-based microbial compositions using the PICRUSt algorithm to make inferences from KEGG annotated databases. Relative signal intensity was normalized by the number of the genes for each indicated metabolic pathway. *, **, *** indicate the difference is at a significant level with *p* < 0.05, *p* < 0.01, and *p* < 0.001, respectively.

## Data Availability

Data is contained within the article or Appendix A.

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
