# Peer review of "Abundant Species Diversity and Essential Functions of Bacterial Communities Associated with Dinoflagellates as Revealed from Metabarcoding Sequencing for Laboratory-Raised Clonal Cultures"

_ijerph, 2022, doi:10.3390/ijerph19084446_

Round 1

Reviewer 1 Report

Abstract

Line 32: What unique features mean here? Please elaborate briefly.

Introduction

Line 54: Please elaborate on "different groups" by providing examples.

Line 76: This seems like a typo "presence of specific interactions their specific interactions". Please clarify the sentence.

Materials and Methods

Line 128: "The identities of all ...". I will suggest the authors to provide the phylogenetic analysis of how the "so said dinoflagellate species" were confirmed. Some description of the method used should also be added the section 2.1. A tree can be provided as supplementary.

Line 133: "Also, 84 of the 144 strains....". It appears these 84 were labeled as "Anti" since they were cultured with antibiotics and the remaining 60 strains were not. It is a bit unclear if a control (no antibiotics) was setup for the same 84 species. A lot of downstream results tend to compare the Anti vs N-Anti and thus, the role of antibiotics on the same strain remains elusive or unclear to me. Please clarify this point. 

Line 146: "approximately 104~105 cells.."This looks like a typo, please correct to 104~105.

Results

Line 215:"accession number PRJNA771505 ....." The deposited data are not accessible and thus cannot be verified. The encourage authors to open the data.

Line 224:"4,738 effective prokaryotic features were further assigned...".However, in line 219, 4832 features were generated. Can the authors confirm these numbers?

Line 280: It was unclear as to why LEfSe and the LDA was meant for. Please elaborate on this briefly.

Line 363: As mentioned before (in method section), this comparison of bacterial communities between 84 Anti strains vs 60 N-Anti strain appears weak. Authors needs to elaborate on this aspect of comparison.

Discussion

This section was well written but some data discussed (e.g. cellulose utilization or methylotrophs, etc) were not described in the results section.

Line 531: understand -> understanding of

Figures

Figure 1: The legends of this figure is very hard to read. Please improve the quality.

Figure 7: Please improve the font of the x-axis of these plots.

Supplementary Fig. S3: Legends, x-y axis needs to be improved.

Author Response

Response to Reviewer 1 Comments

Point 1: Abstract: Line 32: What unique features mean here? Please elaborate briefly.

Response 1: The “unique features” here means that the features were detected only in dinoflagellates, but not in non-dinoflagellates

Point 2: Line 54: Please elaborate on "different groups" by providing examples.

Response 2: Tight associations between microalgae and bacteria have resulted in the evolution of a complex network of cross-kingdom interactions and a fine specialisation of different groups. These interactions are mediated by diverse molecules and recognition mechanisms. For examples, Bacteria belonging to the Roseobacter clade (such as Phaeobacter inhibens and Dinoroseobacter shibae) and Halomonas species are able to provide vitamins (or vitamin precursors) with algae that cannot synthesise them de novo (Croft et al. 2005; Wienhausen et al. 2017). Phaeobacter inhibens and Sulfitobacter can supply auxins and ammonium to diatoms in exchange for the amino acid tryptophan (Amin et al. 2015; Segev et al. 2016). Ruegeria pomeroyi is able to detect different sulfuric compounds released by microalgae and react both by realising auxins that sustain algal growth and quorum-sensing molecules that promotes bacterial proliferation (Johnson et al. 2016).

References:

Amin SA, Parker MS, Armbrust EV. 2012. Interactions between diatoms and bacteria. Microbiology and Molecular Biology Reviews 76: 667-684

Croft MT, Lawrence AD, Raux-Deery E, Warren MJ, Smith AG. 2005. Algae acquire vitamin B12 through a symbiotic relationship with bacteria. Nature 438: 90-93

Johnson WM, Kido Soule MC, Kujawinski EB. 2016. Evidence for quorum sensing and differential metabolite production by a marine bacterium in response to DMSP. ISME Journal 10: 2304–2316.

Segev E, Wyche TP, Kim KH, Petersen J, Ellebrandt C, Vlamakis H, Barteneva N, Paulson JN, Chai L, Clardy J et al. 2016. Dynamic metabolic exchange governs a marine algal-bacterial interaction. eLife 5: 28.

Wienhausen G, Noriega-Ortega BE, Niggemann J, Dittmar T, Simon M. 2017. The exometabolome of two model strains of the Roseobacter group: a marketplace of microbial metabolites. Frontiers in Microbiology 8: 1985

Point 3: Line 76: This seems like a typo "presence of specific interactions their specific interactions". Please clarify the sentence.

Response 3: Thank you for your reminding. The correction has been made as to read now: “Extensive studies have demonstrated specific combinations of algae/bacteria in phycospheres, suggesting the presence of specific interactions (Lines 75-76)”.

Point 4: Line 128: "The identities of all ...". I will suggest the authors to provide the phylogenetic analysis of how the "so said dinoflagellate species" were confirmed. Some description of the method used should also be added the section 2.1. A tree can be provided as supplementary.

Response 4: We completely understand the reviewer's concerns and highly appreciate the reviewer's valuable suggestions. The identities of cultures used in our study were confirmed with multiple lines of evidences, including microscopic observation and confirmation, sequencing the partial LSU rRNA gene (~1400 bp, covering D1-D6 domains), and using the metabarcoding of LSU-PCR amplicons. Among them, all of the 144 cultures were confirmed by using high-throughput amplicon sequencing of 28S rRNA gene. A pair of primers (forward LSU335 5'-ACCGATAGCA(G)AACAAGTA-3' and reverse LSU714 5'-TCCTTGGTCCGTGTTTCA-3') were used to target the highly variable D2 domain and parts of the more conservative D1 and D3 domains of the 28S rRNA gene for eukaryotes. In order to avoid misunderstanding, the sentence in Lines 126-129 of main text has been slightly modified as to read now: “The identities of all cultures were confirmed by high-throughput amplicon sequencing of 28S rRNA gene (~500 bp, covering the highly variable D2 domain and parts of the more conservative D1 and D3 domains).”

Point 5: Line 133: "Also, 84 of the 144 strains....". It appears these 84 were labeled as "Anti" since they were cultured with antibiotics and the remaining 60 strains were not. It is a bit unclear if a control (no antibiotics) was setup for the same 84 species. A lot of downstream results tend to compare the Anti vs N-Anti and thus, the role of antibiotics on the same strain remains elusive or unclear to me. Please clarify this point.

Point 10: Line 363: As mentioned before (in method section), this comparison of bacterial communities between 84 Anti strains vs 60 N-Anti strain appears weak. Authors needs to elaborate on this aspect of comparison.

Response: Since Point 5 and 10 concern the similar issue, they are responded together. We understood the reviewer's concern and appreciated reviewer's suggestion. However, since the purpose of antibiotics mixture was to discourage the growth of bacteria after we established these algal cultures, no backup was set for “without antibiotics addition” as control. Therefore, it was practically impossible to have parallel results for the same set of strains that were not added antibiotics.

Point 6: Line 146: "approximately 104~105 cells.." This looks like a typo, please correct to 104~105.

Response 6: Thank you for your kind reminding. The correction has been made.

Point 7: Line 215:"accession number PRJNA771505 ....." The deposited data are not accessible and thus cannot be verified. The encourage authors to open the data.

Response 7: In response to the reviewer’s concerns, we have requested the NCBI Short Read Archive database to release our data set (PRJNA771505) in advance via e-mail.

Point 8: Line 224:"4,738 effective prokaryotic features were further assigned...".However, in line 219, 4832 features were generated. Can the authors confirm these numbers?

Response 8: Thanks for reviewer's careful reading. The correction has been made, it should be “"4,738”.

Point 9: Line 280: It was unclear as to why LEfSe and the LDA was meant for. Please elaborate on this briefly.

Response 9:  We understood the reviewer's concern. In our study, the LDA (Linear discriminant analysis) effect size analysis (LEfSe) was performed to determine the observed features that most likely to explain differences between samples by coupling standard tests for statistical significance with additional tests encoding biological consistency and effect relevance. This information has been integrated into Lines 193-197 as to read: “The LDA (Linear discriminant analysis) effect size analysis (LEfSe) was performed using the OmicStudio tools to determine the observed features that most likely to explain differences between samples by coupling standard tests for statistical significance with additional tests encoding biological consistency and effect relevance.”

Point 11: Discussion: This section was well written but some data discussed (e.g. cellulose utilization or methylotrophs, etc) were not described in the results section.

Response 11:  In response to the reviewer's concern, we have added related information in the results section, which reads as follows: “Differential function prediction at KO (KEGG Orthology) assignments also revealed members being potentially involved in methanogenesis, including methenyltetrahydrometh-anopterin cyclohydrolase, methylene-tetrahydromethanopterin dehydrogenase, formylmethanofuran-tetrahydromethanopterin formyltransferase, tetrahydrometh-anopterin hydrolyase, formylmethanofuran dehydrogenase subunit A, B, C, were predominantly higher in DINO group than those in N-DINO group (Supplementary Figure S5) (Lines 312-319)” and “Furthermore, we found predominant enrichment of KOs pertaining to the transport system of cellobiose in Athecate group than those in Thecate group, which included cellobiose transport system substrate-binding protein, cellobiose transport system permease protein cebF and cellobiose transport system permease protein cebG (Supplementary Figure S8) (Lines 447-451).”.

Point 12: Line 531: understand -> understanding of

Response 12:  The words have been corrected.

Point 13: Figure 1: The legends of this figure is very hard to read. Please improve the quality.

Response 13:  The new Figures 1 with higher resolution and front size have been provided.

Point 14: Figure 7: Please improve the font of the x-axis of these plots.

Response 14:  The new Figure 7 with higher resolution and front size has been provided in the main text.

Point 15: Supplementary Fig. S3: Legends, x-y axis needs to be improved.

Response 15:  New Supplementary Fig. S3 with higher resolution and front size has been updated.

Reviewer 2 Report

In this study, the authors detail their work on investigating of species diversity and essential functions of bacterial communities associated with dinoflagellates.

The manuscript is very clearly written, with very detailed methodology, and the results are all also clearly presented and thus is an interesting attempt on describing of diversity and functions of algae-associated bacterial community.

specific remarks:

Introduction

Line 76 it seems the phase “specific interactions” has been repeated

Results

General: authors provided very detailed description of their results additionally illustrated with multiple figures. However, their descriptions often gets to convoluted with too many details presented at once, making this pard of the manuscript especially hard to read. Figures also are often unreadable due too low resolution/small font.

Figure 1. figure should be provided in higher resolution, because at this point it is completely unreadable.

Figure 5. as above.

Line 318 “were grouped”?

Line 318-326 This paragraph seems to be written in really convoluted way, I suggest authors rethink what they are trying to convey here.

The referee

Author Response

Response to Reviewer 2 Comments

Comments and Suggestions for Authors: In this study, the authors detail their work on investigating of species diversity and essential functions of bacterial communities associated with dinoflagellates. The manuscript is very clearly written, with very detailed methodology, and the results are all also clearly presented and thus is an interesting attempt on describing of diversity and functions of algae-associated bacterial community.

Response: We sincerely appreciate the reviewer's appreciation of our work. All the raised issues have been addressed accordingly, as highlighted with “Track Changes” in the revised manuscript and response below.

Specific remarks:

Point 1: Introduction: Line 76 it seems the phase “specific interactions” has been repeated.

Response 1: Thank you for your kind reminding. The mistake has been corrected.

Point 2: Results: authors provided very detailed description of their results additionally illustrated with multiple figures. However, their descriptions often gets to convoluted with too many details presented at once, making this part of the manuscript especially hard to read. Figures also are often unreadable due too low resolution/small font.

Response 2: The reviewer's positive comments are highly appreciated. We have updated all the 20 figures (10 in the main text and the other 10 in the Supplementary materials) with higher resolution and improve the font size in the figures in the revision.

Point 3: Figure 1. figure should be provided in higher resolution, because at this point it is completely unreadable. Figure 5. as above.

Response 3: The new figures with higher resolution and front size have been provided.

Point 4: Line 318 “were grouped”?

Response 4: The changed has been made.

Point 5: Line 318-326: This paragraph seems to be written in really convoluted way, I suggest authors rethink what they are trying to convey here.

Response 5: Thanks for the reviewer’s suggestions. We truly hope that the revised version of the entire paragraph as shown below would be better for readers to understand:  “The 130 dinoflagellates were grouping into Thecate (37 strains of the armored) and Athecate (93 strains of the naked) groups. Overall, the two groups shared 716 common bacterial features (17.2 %, as total of 4269 OTUs), while Thecate and Athecate groups had 615 and 2938 unique features, respectively (Figure 2b). At the phylum level, significantly higher relative abundance of Cyanobacteria, Patescibacteria and Actinobacteria was found in Athecate group than in Thecate group; whereas more Verrucomicrobia and Deferribacteres members were present in Thecate group (Figure 7a). Subsequently, bacterial classes of Oxyphotobacteria, Planctomycetacia, Negativicutes, Saccharimonadia, and Actinobacteria exhibited higher abundance in Athecate group, whereas Verrucomicrobiae, Deltaproteobacteria, and Deferribacteres were significantly higher in Thecate group (Figure 7b). At genus level, the Athecate bacterial communities were notably enriched with Oxyphotobacteria, Lautropia, Nonlabens, Mesorhizobium, Klebsiella, Methylotenera, Reichenbachiella, Porphyromonas, Roseovarius, Bacteroides, Blastopirellula, unclassified Actinobacteria and Gammaproteobacteria members (Supplementary Figure S7). In contrast, Athecate group had higher relative abundance in Coprococcus, Lachnospiraceae UCG-006, Tyzzerella, Labrenzia, Roseburia, Akkermansia, Thalassobius, Duncaniella, Lachnospiraceae NK4A136 group, as well as those unclassified Phycisphaeraceae, Nannocystaceae, Muribaculaceae, and Rhodobacterales genera (Supplementary Figure S7). Nonetheless, no significant difference in bacterial communities between Thecate and Athecate groups could be solved by alpha or beta diversity analyses (Supplementary Figure S6)”.

Reviewer 3 Report

Dear editor,

            Thank you for the invitation to read this paper.

            I hereby recommend the acceptance of the manuscript “Abundant species diversity and essential functions of bacterial communities associated with dinoflagellates as revealed from metabarcoding sequencing for more than 100 laboratory-raised clonal cultures, with a possible enrichment of methylotrophs”.

            The description of the phycosphere of 144 dinoflagellates and related algal strains was accessed. The data was analyzed in various statistical methods and thoroughly explored.

            The work is very interesting and thoroughly documented. Comments and suggestions I have added directly to the PDF using the tools and comment function.

            Content to be improved: figures are really small and deserve a higher size, so the reader can explore them accordingly.

           For the supplementary file, Table S1 the stain number CCMA262 P. minimum and CCMA127 A. mimutum.  Does this information need to be corrected?  Authors should check the S1 Table regarding the strains numbers and names.

            The study’s conclusions are supported by the data, and the information is presented in a clear, credible way. The discussion is thorough and well composed, and the reference list is adequate.

            I enthusiastically recommend this manuscript for publication, considering minor revisions.

Author Response

Response to Reviewer 3 Comments

Comments and Suggestions for Authors:  I hereby recommend the acceptance of the manuscript “Abundant species diversity and essential functions of bacterial communities associated with dinoflagellates as revealed from metabarcoding sequencing for more than 100 laboratory-raised clonal cultures, with a possible enrichment of methylotrophs”. The description of the phycosphere of 144 dinoflagellates and related algal strains was accessed. The data was analyzed in various statistical methods and thoroughly explored. The work is very interesting and thoroughly documented. Comments and suggestions I have added directly to the PDF using the tools and comment function.  Content to be improved: figures are really small and deserve a higher size, so the reader can explore them accordingly. The study’s conclusions are supported by the data, and the information is presented in a clear, credible way. The discussion is thorough and well composed, and the reference list is adequate. I enthusiastically recommend this manuscript for publication, considering minor revisions.

Response: Thank you for your positive comments on our manuscript and valuable suggestions to improve the quality of our manuscript. We have updated all the 20 figures (10 in the main text and the other 10 in the Supplementary materials) with higher resolution and improve the font size in the figures in the revision. All the raised issues have been addressed accordingly, as highlighted by using the “Track Changes” in the revised manuscript and response below.

Minor remarks:

Point 1: For the supplementary file, Table S1 the stain number CCMA262 P. minimum and CCMA127 A. mimutum.  Does this information need to be corrected?  Authors should check the S1 Table regarding the strains numbers and names.

Response 1: We appreciate the reviewer's careful reading and kind reminding. We have checked the Table S1 carefully and made corrections in the revision.

Point 2: Title: Don't you think that the title is quite big?

Response 2: Thanks for the reviewer’s suggestions. The title has been modified as follows: “Abundant species diversity and essential functions of bacterial communities associated with dinoflagellates as revealed from metabarcoding sequencing for laboratory-raised clonal cultures”.

Point 3: “suggesting the presence of specific interactions their specific interactions”. Please verify if it is right.

Response 3: Thank you for your reminding. The correction has been made as to read now: “Extensive studies have demonstrated specific combinations of algae/bacteria in phycospheres, suggesting the presence of specific interactions (Lines 75-76)”.

Point 4: Figure 1 is not clear for the reader. It should be much bigger.

Response 4: We appreciate the reviewer's suggestions. The new figure with higher resolution and front size has been provided.

Round 2

Reviewer 1 Report

I thank the authors for providing clarification of several of the points raised before. Most of them have been answered satisfactorily and some additional sections were further elaborated, which has helped a lot. I thank the authors for this.

There are still few minor issues that need to be addressed:

  1. Page 6, Line 239 (Figure 1): The x-axis still remains illegible; there is too much overlap between the IDs on the axis. Maybe authors can include a sentence on the figure legend to point readers to a refer to the detailed list of organisms in the supplementary table.
  2. Line 217: Data from PRJNA771505 still remains inaccessible, although authors have made an effort for the release from NCBI. 
